# Advanced Glycation End Products and Diabetes Mellitus: Mechanisms and Perspectives

**DOI:** 10.3390/biom12040542

**Published:** 2022-04-04

**Authors:** Mariyam Khalid, Georg Petroianu, Abdu Adem

**Affiliations:** Department of Pharmacology and Therapeutics, College of Medicine and Health Sciences, Khalifa University, Abu Dhabi P.O. Box 127788, United Arab Emirates; mariyam.khalid@ku.ac.ae (M.K.); georg.petroianu@ku.ac.ae (G.P.)

**Keywords:** hyperglycemia, type 2 diabetes mellitus, advanced glycation end products (AGEs), receptor for advanced glycation end products (RAGE), pancreatic beta cells, diabetic complications

## Abstract

Persistent hyperglycemic state in type 2 diabetes mellitus leads to the initiation and progression of non-enzymatic glycation reaction with proteins and lipids and nucleic acids. Glycation reaction leads to the generation of a heterogeneous group of chemical moieties known as advanced glycated end products (AGEs), which play a central role in the pathophysiology of diabetic complications. The engagement of AGEs with its chief cellular receptor, RAGE, activates a myriad of signaling pathways such as MAPK/ERK, TGF-β, JNK, and NF-κB, leading to enhanced oxidative stress and inflammation. The downstream consequences of the AGEs/RAGE axis involve compromised insulin signaling, perturbation of metabolic homeostasis, RAGE-induced pancreatic beta cell toxicity, and epigenetic modifications. The AGEs/RAGE signaling instigated modulation of gene transcription is profoundly associated with the progression of type 2 diabetes mellitus and pathogenesis of diabetic complications. In this review, we will summarize the exogenous and endogenous sources of AGEs, their role in metabolic dysfunction, and current understandings of AGEs/RAGE signaling cascade. The focus of this review is to recapitulate the role of the AGEs/RAGE axis in the pathogenesis of type 2 diabetes mellitus and its associated complications. Furthermore, we present an overview of future perspectives to offer new therapeutic interventions to intervene with the AGEs/RAGE signaling pathway and to slow down the progression of diabetes-related complications.

## 1. Introduction

Type 2 diabetes mellitus is a multifactorial metabolic disorder, broadly categorized by a single diagnostic criterion, hyperglycemia. It is considered one of the most common diseases of the 21st century, increasing at an alarming rate worldwide and predicted to affect 693 million adults by 2045 [1,2,3]. Microvascular and macrovascular complications are the leading cause of morbidity and mortality in people with type 2 diabetes mellitus [4,5]. Although the pathophysiology of type 2 diabetes mellitus is complex, advanced glycation end products (AGEs) are identified as key players in the progression of diabetes and diabetes-induced complications. Uncontrolled hyperglycemia is considered to be the principal etiology of diabetic complications, as it leads to the production of AGEs [4,6].

AGEs are produced as a result of a classical Maillard reaction when reducing sugars react non enzymatically with an amino group of proteins, lipids, and nucleic acids through a series of reactions forming a Schiff base, followed by an Amadori rearrangement and subsequent oxidative modifications (glycoxidation) to produce AGEs [7,8,9]. The glycation process is complex and slow but depends on the availability of substrates and causes spontaneous damage to proteins in physiological systems [10]. Therefore, during normal physiological conditions, there is moderate production of AGEs, but it is markedly accelerated under persistent hyperglycemic conditions due to increased glucose availability [6,8,10]. Previously, advanced glycation was considered to affect only long-lived proteins (for example collagen and elastin). However, in chronically elevated hyperglycemia, advanced glycation can also modify the short-lived proteins, including circulating plasma proteins and lipids. The level of circulatory AGEs is significantly elevated in diabetic patients, specifically with impaired renal clearance [10,11,12].

The most compelling explanation for the deleterious effects of AGEs is the irreversible damage to the structural and functional integrity of proteins through intermolecular and intramolecular crosslinks [13,14,15]. Adjacent AGE molecules have the capability to crosslink with each other and with certain proteins that alter the structure and interfere with the functional properties of proteins. Due to this covalent cross-link formation, the biologically active proteins and enzymes are deactivated [8,16], become resistant to proteolytic digestion [17,18], provide catalytic sites for ROS formation [19], lead to pro-inflammatory events [6], alter intracellular signaling and contribute to several metabolic and biochemical perturbations [10,17,18,20]. Moreover, the interaction of AGEs with a wide range of cell surface receptors elicits numerous cell-mediated pathophysiological responses that might provide the mechanistic links between the onset of diabetes mellitus and AGEs accumulation. Upon binding to their cognate receptor, AGEs trigger the activation of multiple signals that can directly affect cellular function, and metabolism through upregulation of inflammation and oxidative stress [21].

In this brief review, we will summarize the variety of sources and processes of AGEs formation as well as their role in metabolic dysfunction. The focus of this review is to recapitulate the current understanding of the basic molecular mechanisms triggered by AGEs that contributes to the pathogenesis of type 2 diabetes mellitus and its associated complications. These observations will provide a future perspective to offer new therapeutic interventions to counteract AGEs related damage in type 2 diabetes mellitus.

## 2. Sources of AGEs

AGEs accumulation can be through either endogenous or exogenous sources [22]. The exogenous AGEs are present in a wide variety of food items [8]. Cigarette smoke also contains glycation products that are highly reactive and act as a precursor of AGEs formation [22,23]. Exogenous AGEs are found in high levels in the modern Western diet. The thermal processing of food, specifically by using dry heat technology in cooking such as frying, grilling, baking or barbecuing, results in substantial AGEs formation [23]. Food processing to enhance conservation and safety and to improve flavor and appearance also lead to the generation of diverse food-derived AGEs known as glycotoxins. [10,14,24]. The heterogeneity of AGEs depends upon the particular structure of protein-bound AGEs, which defines its novel modification to a particular native protein. The classification of AGEs into protein-bound, peptide-bound or free AGEs predict their rate of absorption or pinpoint potential transporters [25,26]. Depending on the chemical characteristics of dietary AGEs, only 10–30 percent of ingested AGEs are absorbed into the systemic circulation [25]. The mechanism of gastrointestinal metabolism and absorption of food-derived AGEs is yet to be fully elucidated [14,22]. However, it mostly depends upon the molecular weight of the products of protein hydrolysis and the type of required peptide transporters [27]. The unabsorbed AGEs that are delivered to the colon can affect gut microbiota homeostasis and induce an inflammatory response to modify gut integrity [22,23]. This local inflammatory response is associated with increased systemic inflammatory cytokines responsible for compromised glucose control [22,28]. It is now well documented that exogenous AGEs contribute significantly to the body’s AGEs pool [22,25,28].

The predominant process of endogenous AGEs formation is through the complex, multistage glycation process Maillard reaction [9]. This nonenzymatic process of glycation is accelerated in hyperglycemic conditions, such as in diabetes mellitus. Maillard reaction generates highly reactive numerous intermediate carbonyl precursors of AGEs [20,29]. Other than this nonenzymatic reaction, dicarbonyls, also known as α-oxoaldehydes, are generated endogenously through glucose autoxidation, polyol pathway, and lipid peroxidation, as described in Figure 1. Previous studies have shown that under sustained hyperglycemic conditions as in type 2 diabetes mellitus, glucose toxicity is induced by increased glucose flux through the glycolytic pathway [30]. The consequence of continuous glycolysis results in dihydroxyacetone phosphate (DHAP) accumulation due to the decline in a crucial glycolytic enzyme, triose phosphate isomerase (TPI), activity. Due to insufficient activity of the enzyme, the interconversion of DHAP and glyceraldehyde-3-phosphate (GAP) catalyzed by TPI is not efficiently possible, thereby leading to the spontaneous formation of the highly reactive bicarbonyl, methylglyoxal [31,32].

An increase in intracellular glucose levels is associated with oxidative stress, autoxidation of glucose, and channels glucose towards the polyol pathway [27,33,34,35]. Lipid peroxidation is also increased in diabetes to produce advanced lipid peroxidation end products (ALEs). Polyunsaturated fatty acids are oxidized to produce reactive carbonyl species such as malondialdehyde and methylglyoxal, leading to the synthesis of the well-characterized ALEs [21]. The dicarbonyl intermediates are the important focal point of endogenous AGEs formation. Additionally, dicarbonyl stress is also generated by ketone metabolism in uncontrolled hyperglycemia [21,25,36].

## 3. Pathophysiology of AGEs/RAGE in Diabetes Mellitus

Two primary mechanisms are involved in the AGEs-induced pathophysiology of diabetes mellitus. The AGEs exert their deleterious effects, either directly by trapping and cross-linking of proteins, or indirectly by binding to the cell surface receptor [22,29]. AGEs can signal through several receptors, however AGEs interactions with AGEs receptors and their role in mediating cellular responses are yet to be fully elucidated [12,25]. AGEs can modulate cellular functions through binding with Toll-like receptors, scavenger receptors, G-protein-coupled receptors, and pattern recognition receptors [12,37]. Among these, the most important cell surface receptor for AGEs is the receptor for advanced glycation end products (RAGE). It is a member of the immunoglobulin superfamily, which was initially identified and named for its ability to bind with AGEs [38,39]. One of the salient features of the receptor is its capability to bind a broad repertoire of ligands [39,40]. RAGE recognizes three-dimensional structures rather than specific amino acid sequences. This multiligand receptor is considered a pattern-recognition receptor because of its ability to identify the structure of ligand recognition sites [38,39,41].

The human *RAGE* gene is located on chromosome 6 close to major histocompatibility complex III (MHC class III), which indicates its involvement in immune responses [42,43]. The resulting transcribed mRNA translates into a protein of 404 amino acids with a mass of 45–55 kDa [16,39]. Full-length RAGE (fl-RAGE) is comprised of three domains, an extracellular domain (*N*-terminal V-type domain and two C-type (C1 and C2 immunoglobulin domains), a hydrophobic transmembrane domain, and a highly-charged amino acid cytosolic domain [38,39]. The V-type domain from the extracellular region, in particular, interacts with the potential extracellular ligands, while the cytoplasmic tail is critical for intracellular signaling and serves as a scaffolding for the initiation of signal transduction [44]. In addition to the fl-RAGE, recently, numerous naturally occurring RAGE protein isoforms have also been described. The RAGE primary transcript undergoes alternative splicing and proteolytic cleavage of fl- RAGE under the control of yet-unknown pathways to produce truncated RAGE iso-forms [42,43,45]. The *N*-terminal truncated lacks the ligand-binding domain and is unable to engage glycated end products. The C-terminal truncation majorly forms a pool of soluble RAGE (sRAGE) including endogenous secretory RAGE (esRAGE) generated from alternative splicing and cleaved RAGE (cRAGE) derived from the proteolysis of membrane-bound fl-RAGE by metalloproteases [42,46]. The sRAGE lacks a transmembrane domain and functions as decoy receptor as it releases into the extracellular space and interacts with RAGE ligands preventing membrane-bound fl-RAGE/ligands cell signaling, as well as altering the generation and maturation of potential RAGE ligands. The dominant-negative RAGE (dnRAGE) lacks the cytosolic tail, thus blocking the activation and signaling of fl-RAGE [39,44]. Presumably, both the sRAGE and dnRAGE interfere with fl-RAGE receptor toxic signal transduction and play an antagonistic role in AGEs/RAGE signal transduction [38,46]. The fl-RAGE isoform is the most prevalent RAGE isoform and is present in numerous cell types throughout the body [25,39]. The membrane-bound fl-RAGE is responsible for intracellular RAGE signaling in response to extracellular ligands that lead to activation of the proinflammatory events [22,39]. The AGEs mediated RAGE activation promotes upregulation of RAGE receptor expression. This positive feedback loop indicates that ligand stimulated RAGE receptor acts as a propagation and perpetuation factor [47,48].

During chronic diabetes, persistent hyperglycemia leads to elevated levels of AGEs in the bloodstream, which by engagement to RAGE induces an array of signaling events. AGEs/RAGE interaction triggers a variety of downstream effectors including mitogen-activated protein kinase (MAPK), p38, stress-activated protein kinase/c-Jun *N*-terminal kinase (SAPK/JNK), Ras-mediated extracellular signal-regulated kinase (ERK1/2), and Janus kinase signal transducer and activator of transcription (JAK/STAT) pathway that in turn will lead to sustained activation transcription factors such as NF-κB, STAT3, HIF-1α, and AP-1 [25,49,50].

The activation of JNK promotes the phosphorylation of insulin receptor substrate (IRS-1) at serine residues that leads to negative regulation of insulin signal transduction and induces insulin resistance [51]. The phosphorylation of serine residues in the insulin receptor (IR) and IRS-1 molecule results in diminished enzymatic activity in the phosphatidylinositol 3-kinase/protein kinase B (PI3K/Akt) pathway. The RAGE transduction induced IκB kinase (IKKβ) activation promotes phosphorylation and ubiquitination mediated proteasomal degradation of inhibitor of NF-κB (Iκβ) proteins, thus releasing NF-κB. The activated master transcription factor NF-κB translocates to the nucleus and upregulates the expression of various inflammatory cytokines (IL-1β, IL-6, TNFα) that can cause insulin resistance [51]. The AGEs/RAGE signaling, as well as increased inflammation, leads to activation of MAPK, p38, and protein kinase C (PKC). These kinases will mediate insulin resistance directly by downregulating insulin receptor expression, impair IRS-1 tyrosine phosphorylation, promote IRS-1 serine phosphorylation, leading to defective insulin receptor signaling [51,52,53]. Additionally, recent findings have also implicated abnormal activation of the ERK1/2 signaling pathway in diabetes which will influence the upregulation of several diabetogenic factors and promote adipogenesis [54,55]. The increased inflammation further triggers the activation of additional mediators which increases inflammation, as well as activates the signal transducer and activator of transcription 3 (Stat3) [52]. STAT3 induces insulin resistance in muscles by leading to degradation of IRS-1 through the upregulation of F-Box Protein 40 (Fbxo40), a muscle-specific E3 ubiquitin ligase [56]. Under similar conditions of hyperglycemia and AGEs accumulation, the interplay between RAGE-induced cellular dysfunction, protein kinases, and inflammation that lead to sustained activation transcription factors such as NF-κB, STAT3, HIF-1α, and AP-1 further attenuates insulin sensitivity in target cells [52,53,57,58]. The persistent activation of NF-κB, in addition to the perpetuation of chronic low-grade inflammation, positively regulates RAGE expression by binding to its proximal promoter region [25,59].

Recently, studies showed that RAGE/NF-κB signaling also activates NLRP3 inflammasome formation, which is a critical component of the innate immune system. The NLRP3 inflammasome in response to cellular stress-signals mediates caspases-1 cleavage and contributes to the maturation and secretion of key inflammatory cytokines IL-1β/IL-18 [60,61]. Various human studies establish a correlation between increased NLRP3 expression and insulin resistance [62,63,64]. The overexpression of RAGE also promotes de novo synthesis of NF-κB p65 (REL A), which results in a high level of transcriptionally active NF-κB, overriding the endogenous negative feedback mechanisms [59,65]. The NF-κB p65 directly induces insulin resistance by repressing the transcription of glucose transporter GLUT4 protein, codified by *Slc2a4* gene in skeletal muscles by binding to the *Slc2a4* gene promoter [66,67,68].

Such abiding AGEs/RAGE interaction, elevated levels of NF-κB, PKC, and NLRP3 inflammasome activation transduce ROS generation, via activation of nicotinamide adenine dinucleotide phosphate oxidase (NADPH oxidase) [49]. Increased levels of ROS will overburden the activities of superoxide dismutase (SOD) and catalase and will diminish glutathione stores. This imbalance in the intracellular redox state will result in oxidative stress in the endoplasmic reticulum (ER), which is strongly interconnected to mitochondria through mitochondria-associated ER membranes (MAMs). MAMs through the exchange of metabolites and ions between these two organelles maintain cellular homeostasis. The increased ER stress will cause mitochondrial dysfunction, alter redox homeostasis, play a crucial role in damaging cellular processes and the infrastructure of the cell, and contribute to oxidative stress propagation [2,69,70]. The mitochondrial ROS production subsequently activates abnormal activation of several kinases such as MAPK, ERK, IKK, p38, JNK involved in stress responses, which will trigger the vicious cycle of inflammation and ROS generation [71,72]. A plethora of evidence suggests AGEs/RAGE signaling pathway, NF-κB activation, inflammation, and ROS generation are directly related to the pathogenesis of insulin resistance by increased IRS-1serine phosphorylation and degradation, thus blocking the insulin signaling pathway [68,71,73,74].

## 4. AGEs/RAGE Axis and Pancreatic Beta Cells

Hyperglycemia-induced AGEs load in the pancreas contributes to beta cell toxicity via activation of inflammatory cascades and oxidative stress [75]. High levels of AGEs upregulate RAGE expression in pancreatic islets, as observed in several studies [76]. The AGEs/RAGE axis triggers intracellular signal transduction and activates NF-κB transcription, resulting in chronic inflammation, mitochondrial dysfunction, beta cell impairment, and apoptosis [67,76,77].

Islet amyloid polypeptide (IAPP) is another major factor that contributes to pancreatic beta cell death in diabetes [78]. Substantial evidence reveals that RAGE selectively binds with toxic IAPP intermediates and transduces intracellular signals that lead to NADPH oxidase-mediated ROS generation, induce cellular stress, inflammation, and play a key role in islet amyloidosis–induced beta cell proteotoxicity [79,80] as elaborated in Figure 2. The aberrant accumulation of these pathological aggregates leads to decreased beta cell mass and increased beta cell apoptosis, as observed in chronic diabetes [81,82]. Several studies showed that inhibition of the RAGE-by-RAGE neutralizing antibody or through the administration of sRAGE in either the in vitro or in vivo model, preserved beta cell morphology and blocked inflammatory mediators and amyloid formation [77,79,80]. Further research on AGEs/RAGE axis contributing to IAPP-induced islet beta cell toxicity can provide the missing pathological link for the diagnostic criteria and therapeutic intervention for beta cell preservation in chronic diabetes.

## 5. AGEs/RAGE Axis in Diabetic Complications

The engagement of RAGE by AGEs leads to sustained cellular dysfunction, recently termed as “metabolic memory” [83]. The metabolic memory is the long-term influence of previously accumulated AGEs that are capable of maintaining RAGE over-expression, sustained activation of NFκB, prolonged induction of tissue-specific inflammation, initiation and progression of long-term oxidative stress, which is persistent despite the reversal of hyperglycemia [84,85,86]. The phenomenon of this hyperglycemic memory, instigated by the AGEs/RAGE axis, is associated with the pathogenesis of diabetes complications [87,88]. Diabetes-related macrovascular and microvascular complications are responsible for the impaired quality of life, accounting for increased morbidity, disability, mortality, and contributing substantially to healthcare costs [89,90].

Extensive epidemiological data indicate that patients with type 2 diabetes are at higher mortality risk from cardiovascular disease (CVD) compared to non-diabetics, across different races, regions, and sex. The inadequate management of diabetes provokes hyperglycemia-induced cardiovascular events through various mechanisms [89,91]. Evidence supports a direct correlation between the AGEs/RAGE axis, activated signal transduction of MAPK and NFκB cascades and intracellular ROS generation, subsequently leading to the production of several inflammatory and profibrotic factors such as vascular cell adhesion molecule-1 (VCAM-1), intercellular adhesion molecule-1 (ICAM-1), plasminogen activator inhibitor-1 (PAI-1), monocyte chemoattractant protein-1 (MCP-1), and matrix metalloproteinase (MMP)-2 protein [92,93]. Increased expression of these prothrombotic species is involved in arterial stiffness, vascular calcification, and plaque accumulation in atherosclerosis-prone vessels [87,94]. AGEs/RAGE mediates the increase in oxidative stress and enhances the oxidation of low-density lipoprotein (LDL), which is the key player in the pathogenesis of CVD [95]. Oxidized LDL act as a ligand for RAGE, leading to the activation of multiple intracellular pathways such as NF-κB, p38, JNK, and MAPK, augmenting the expression of TGF-β, C-reactive protein (CRP), inflammatory cytokines, and PKC. This leads to increased vascular calcification and hardening of the medial layer of blood vessels, which ultimately contributes to the pathophysiology of CVD [96,97,98,99], as elaborated in Figure 3.

Moreover, the nonenzymatic modifications of collagens and lipoproteins by AGEs in large vessels will result in increased collagen deposition, altering the structural integrity of arteries, disarray of elastic fibers, and the degeneration of smooth muscle tissue, which are key pathogenic factors in arteriosclerosis [100,101]. The accumulation of AGEs on long-lived matrix proteins (collagen, elastin) is associated with AGEs-related crosslinks which will result in increased arterial stiffness, and endothelial dysfunction and disrupt extracellular matrix-cell (ECM) interactions [102,103,104]. This nonenzymatic modification of collagen is not only implicated in CVD but also profoundly involved in nephropathy, inflammatory bowel disease, osteoporosis, neuropathy, and retinopathy [100,105]. The increased arterial stiffness associated with systemic microinflammation can lead to pressure fluctuations in the microvasculature of different organs, specifically kidneys, and may increase the risk of renal failure. [106,107]. According to the WHO global report in 2016, approximately 12–55 percent of incidences of end-stage renal disorders (ESRD) are attributed to type 2 diabetes [108]. The recent cross sectional study also demonstrated high prevalence of chronic kidney disease (CKD) in patients with diabetes with poor glycemic control, persistent albuminuria and reduced estimated glomerular filtration rate (eGFR) [109].

AGEs are metabolic mediators of kidney damage as its correspondent receptor RAGE is expressed by several cell types in the kidney as podocytes [110], tubular epithelial cells, and mesangial cells [84,88]. In uncontrolled hyperglycemia, AGEs accumulation in kidneys is accelerated and it mechanistically activates diverse signal transduction cascades by binding to RAGE [88]. The downstream consequences of AGEs/RAGE interaction are through the activation of NF-κB, MAPK, JNK, and TGFβ, leading to ROS generation. The ROS burden will reduce antioxidant enzymes and cellular glutathione levels, resulting in the up regulation of NADPH oxidase, nitric oxide synthase (NOS), and cyclooxygenase (COX) [88,111,112]. These events will evoke monocyte chemoattractant protein-1 (MCP-1), leading to leukocyte infiltration, over-expression of various cytokines and intracellular adhesion molecules, extracellular matrix accumulation, increased angiotensin II levels, and calcium influx, which will further exacerbate the inflammation [87,88,113,114]. The hallmark of diabetic renal injury includes increased microalbuminuria, renal podocyte injury, podocyte protein accumulation, affected renin-angiotensin system, glomerular and tubular hypertrophy, and kidney fibrosis culminating in gradual loss of kidney architecture and function [102,112,115]. Furthermore, RAGE activation accelerates heparinase secretion, which disintegrates the glomerular filtration barrier by degradation of heparin sulfate, a fundamental part of the glomerular basement membrane (GBM) [102,116]. Several animal and human studies validate the involvement of AGEs/RAGE axis and stimulation of its downstream signaling pathways in the pathophysiology of diabetes and its associated micro-vascular and macro-vascular complications, such as cardiomyopathy, nephropathy, retinopathy, and neurodegeneration [84,86,87,88,114], as described in Figure 3.

The cross-talk between environmental factors and diabetogenic factors such as lifestyles, improper diet, hyperglycemia, inflammation, oxidative stress and AGE/RAGE axis contribute to the etiology of this unique phenomenon of metabolic memory [83,117]. Metabolic memory often translates these molecular alterations to epigenetic aberrations which perpetuate the expression of pathological genes associated with the progression of diabetic complications [118,119]. The epigenetic mechanisms include DNA methylation associated with transcriptional silencing, histone post-translational modifications which regulate gene expression by recruiting chromatin remodelers, as well as transcription co-activators and co-repressors, and non-coding RNAs in the complex interplay between genes and the environment. The epigenetic change in histone alterations (methylation/acetylation) serves as a link between the pathogenesis of diabetes, metabolic memory, and diabetes-related complications, as it is implicated in the upregulation of NFκB-p65 gene expression and inflammation [118,119,120].

## 6. Future Perspectives

AGEs constitute a complex heterogeneous group of compounds that can be produced endogenously or taken up as dietary AGEs [18,27]. The modern heat-processed, highly palatable, and appetite-enhancing, AGEs-rich diet shift over the past half-century has contributed to increased oxidative stress and inflammation, which is linked to the recent epidemic of diabetes [23,24]. Uncontrolled continued exposure to hyperglycemic state is a precursor to AGE formation and accumulation in diabetes mellitus. This non-enzymatic glycation and cross-linking of protein, lipids, and nucleic acids modifications alter the structural integrity and function of these macromolecules [36,87]. The deleterious effects of AGEs are underpinned by their ability to trigger the release of pro-inflammatory molecules, impair mitochondrial oxidative stress, activate AGEs/RAGE signaling cascade, and trigger transcription factors that upregulate the expression of genes which have potential roles in the pathogenesis of diabetic complications [13,25,41]. This is the reason that the role of AGEs in diabetic complications is one of the most promising research areas today, as it triggers many important biochemical mechanisms which are central to the pathogenesis of type 2 diabetes mellitus and its associated complications. Recent studies assessed the concentration of different circulating AGEs in the serum samples of patients with chronic type 2 diabetes treated with oral hypoglycemics and associated it with the pathobiology of diabetes and severity of diabetic complications [16,121].

Reducing dietary AGEs burden through lifestyle modification to modulate AGEs/RAGE axis for better health is one of the most important interventions and practical ways to reduce the load of exogenous AGEs. The lifestyle modifications include dietary AGE restriction, moderate exercise, and constraint cigarette smoking. In this regard, one of the most challenging and crucial tasks is the development of a specific, sensitive, cost-effective, and universally accepted method for the measurement of circulatory AGEs, as sophisticated and expensive laboratory techniques cannot be used extensively to keep a check on the body’s AGEs pool. Moreover, in order to design logical intervention, further research regarding the nature of different types of AGEs and their associated cellular response is needed.

The interaction of AGEs/RAGE results in perturbations of a variety of signaling pathways which causes oxidative stress, and activation of NF-κB, TGF-β, MAPK signaling resulting in the modulation of gene transcription and generation of pro-inflammatory cytokines [12,67,122]. Thus, pharmacological intervention interfering either directly or indirectly with the AGEs/RAGE axis can prove to be a valid therapeutic approach. Among all the isoforms of RAGE, only fl-RAGE is responsible for RAGE-induced pathological intracellular signals [38,44,114]. The soluble form of RAGE (sRAGE and esRAGE) is devoid of the cytoplasmic domain and capable of suppressing the RAGE signaling, although esRAGE function is far more complex than just a putative decoy against RAGE signaling [44,46]. The s-RAGE plasma level is used as a biomarker for RAGE-mediated diabetic complications [123]. As RAGE is responsible for the activation of a myriad of signaling pathways, RAGE blockade is another potential treatment option. Several in vivo animal studies showed that genetic deletion or pharmacological blockade of the RAGE prevents the progression of diabetic complications. Blockade of RAGE using sRAGE in diabetic mice either induced by streptozotocin or genetic db/db type 2 diabetes model attenuated inflammation and vascular complications [114,124]. Deletion of RAGE prevented diabetic nephropathy in OVE26 type 1 mouse, a model of progressive glomerulosclerosis, and a decline in renal function [125]. Furthermore, animal studies confirmed the successful use of humanized anti-RAGE monoclonal antibodies to block RAGE-mediated sepsis to improve the survival rate in BALB/c mice [65,126].

Recently, RAGE inhibitor PF-04494700, the small nontoxic molecule, entered a phase 2 clinical trial to antagonize amyloid-β load in Alzheimer’s disease [127]. Another specific RAGE antagonist FPS-ZM1 showed to be neuroprotective against ischemic brain injury in a distal middle cerebral artery occlusion (MCAO)-induced rat model. [128]. One other study demonstrated FPS-ZM1 as a high-affinity RAGE-specific blocker that inhibits amyloid-β binding with RAGE in an APP(sw/0) transgenic mouse model of Alzheimer’s disease [129]. These promising outcomes of RAGE inhibitors in neurodegenerative diseases provide the rationale to design a clinical trial for the evaluation of these specific RAGE blockers in patients with chronic diabetes, although the potential consequence of antagonizing RAGE needs to be further elucidated.

The other receptors of AGEs such as AGER1 and AGER3/galectin3 counteract the effects of AGEs and show opposite effects compared to RAGE [130,131]. AGER1 encoded by the gene Dolichyl-Diphosphooligosaccharide-Protein Glycosyltransferase *(DDOST)* is present in almost all cells and tissues. AGER1 accelerates the uptake and degradation of AGEs by promoting sirtuin1-dependent deacetylation and suppression of NF-κB. Synergism between AGER1 and SIRT1 suppresses AGEs-induced ROS formation, NF-κB, and MAPK activation [12,132]. AGER1 competitively interferes with RAGE and binds AGEs with high affinity due to its long extracellular tail [12]. Overexpression of AGER1 provides a shield to SIRT1 and blocks transcription of inflammatory genes, while downregulation of AGER1 is associated with AGEs and ROS accumulation, as observed in chronic diabetes [12,67]. As AGEs are colocalized with RAGE, increased AGEs accumulation stimulates the upregulation of RAGE expression, mainly through NFκB-p65 activation. NF-κB is a master transcription factor involved in the upregulation of RAGE, inflammatory cytokines, and other genes involved in insulin resistance [62,111]. Inhibition of NFκB, specifically its p65 subunit, may provide an effective treatment option for diabetic complications. So, there is a need to explore novel anti-inflammatory agents able to upregulate AGER1 and suppress NFκB-p65 in diabetic complications with the help of preclinical studies and clinical trials.

Another option for AGEs detoxification is the early degradation of AGEs by chemopreventive enzymes and antioxidants, such as lactoferrin, defensins, and lysozyme circulating in the blood. The most effective of these is lysozyme, a well-characterized, naturally occurring protein with antimicrobial properties [122,133,134]. It possesses a distinctive combination of anti-AGEs properties, accelerates renal AGE clearance, and suppresses intracellular AGEs-mediated signaling. The glyoxalase enzyme system utilizes reduced glutathione, Glyoxalase I (GLOI), and Glyoxalase II (GLOII) for the detoxification of AGEs. Studies in cultured cells and model organisms showed that overexpression of GLO1 associated with the inhibition of AGEs induced oxidative stress, while the downregulation of GLO1 expression is associated with elevated levels of AGEs accumulation [25,122]. The glyoxalase system is present in all cells and can efficiently scavenger dicarbonyl stress. Therapeutic agents, either a synthetic drug such as candesartan [135], or compounds isolated from natural sources such as resveratrol [136], fisetin [137], sulforaphane [138] and cyanidin [139], have shown promising effects to increase GLO1 activity and up-regulate GLO1 expression. The binary combination of natural compounds trans-resveratrol and hesperetin (tRES-HESP) effectively restored GLO-I activity in overweight and obese subjects [140]. The detoxification pathway of DJ-1/Park7 is referred to as DJ-1 family of Maillard deglycases. DJ-1 is a multifunctional stress response protein, although glutathione-independent, functions similar to the glyoxalase system, in reversing glycation [25,141]. The detoxifying enzyme systems described so far can play a pivotal role in maintaining AGEs homeostasis by retarding AGEs formation and may prove instructive in the design of novel therapies against diabetic complications.

Various laboratory and clinical data have shown the significance of urea-based small molecules such as aminoguanidine and metformin having powerful antioxidant properties [142,143,144]. They inhibit the subsequent production of protein bound carbonyl content and AGEs either through their scavenger mechanism of α-dicarbonyl compounds or through an increase in enzymatic detoxification [145,146].

The use of medicinal natural compounds (phytochemicals) having antioxidant and anti-inflammatory properties is another potential option to counteract the downstream consequences of the AGEs/RAGE axis. Several studies have shown the beneficial effects of phytochemicals such as carbonyl scavengers, protein glycation inhibitors [147,148] and free radicals scavengers [149] that result in reduced lipid peroxidation and suppress inflammation and oxidative stress [96,138].

The AGEs/RAGE axis activates a myriad of signaling pathways with various downstream consequences such as oxidative stress, inflammation, perturbation of metabolic homeostasis, and epigenetic modifications [96]. The identification of a combination therapies targeting multiple pathways can prove to be a better option for the intervention of AGEs accumulation and blockade of the AGEs/RAGE axis, rather than targeting single pathways alone. The implementation of novel and targeted integrative approaches such as AGEs inhibitors, AGEs/RAGE blockers, free-radical scavengers, and anti-inflammatory adjunct therapies may prove to be beneficial for the prevention of diabetes-associated pathogenesis. Despite multiple advances in therapeutic interventions, many gaps remain in metabolic memory-related diabetic pathophysiology that need to be further elucidated.

## Figures and Tables

**Figure 1 biomolecules-12-00542-f001:**
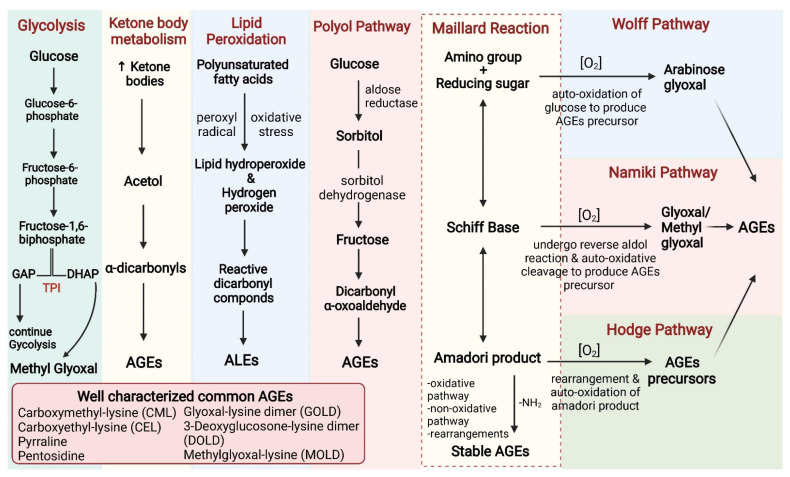
Pathways of AGEs formation. Maillard reaction is the classical pathway of advanced glycation end products (AGEs) formation. Metal catalyzed autooxidation of glucose (Wolff pathway) or reverse aldol reaction and autooxidation of the Schiff base (Namiki pathway) or non-oxidative Amadori product cleavage (Hodge pathway) forms reactive dicarbonyls and AGEs precursors that contribute significantly to AGEs formation. In the Polyol pathway, an excessive amount of glucose is reduced to sorbitol by the enzyme aldol reductase, which is then converted to fructose by sorbitol dehydrogenase. Fructose and its metabolites, being potent glycating agents, lead to the production of AGEs. The lipid peroxidation of polyunsaturated fatty acids leads to the formation of lipid peroxides, which first converts to reactive dicarbonyls and ultimately results in the formation of advanced lipid peroxidation end products (ALEs). The amino acid derived ketone body metabolism also generates AGEs by producing the intermediate AGEs precursor reactive dicarbonyls. The endogenous generation of methylglyoxal via the triose phosphate isomerase reaction of glycolysis is also an important source of AGEs. Some of the important reactive dicarbonyls involved in AGEs formation include methylglyoxal, glyoxal, and 3-deoxyglucosone.

**Figure 2 biomolecules-12-00542-f002:**
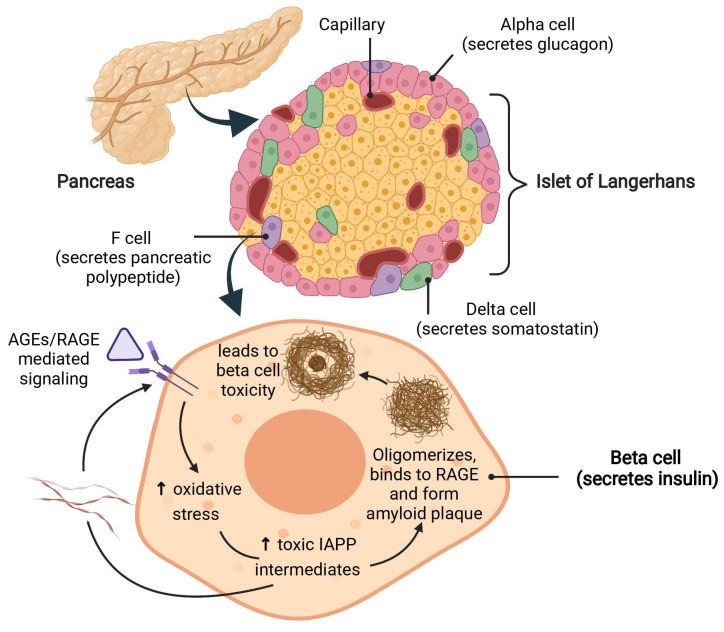
AGEs/RAGE-induced pancreatic beta cell toxicity. The AGEs/RAGE mediated signaling causes enhanced oxidative stress and increased inflammation in pancreatic beta cells. The generation of damaging reactive oxygen species (ROS) during oxidative stress affects the amyloidogenicity of islet amyloid polypeptide (IAPP) and leads to the formation and aggregation of toxic IAPP species. RAGE binds with these toxic IAPP intermediates and leads to the formation of amyloid plaque resulting in pancreatic beta cell toxicity.

**Figure 3 biomolecules-12-00542-f003:**
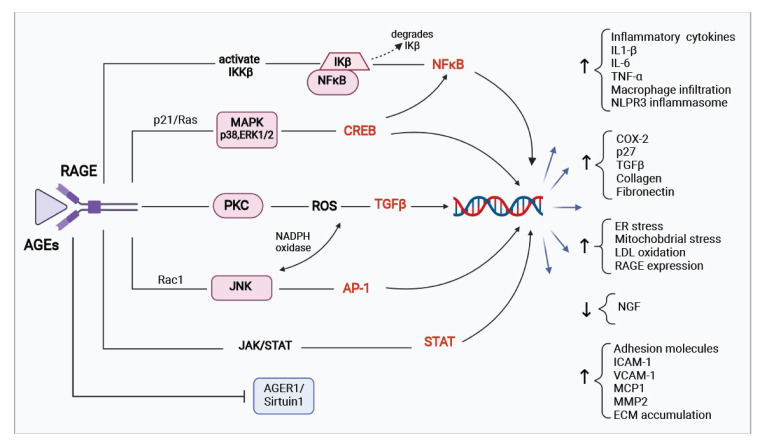
AGEs/RAGE axis and diabetic complications. The AGEs/RAGE interaction triggers several signaling cascades such as IKK/NF-κB, ERK/MAPK, PKC, JNK and JAK/STAT and activates transcription factors such as NFκB, CREB, AP-1, and STAT3, which results in oxidative stress and amplifies inflammatory responses. The AGEs/RAGE axis will stimulate macrophage infiltration, increase gene expression of inflammatory cytokines, adhesion molecules, and ECM proteins. It will result in overexpression of TGF-β, fibronectin, and collagen. The increase in ER stress will lead to enhanced oxidative stress which will further cause inflammatory responses and activate LDL oxidation. The RAGE expression will be upregulated in response to positive feedback by AGEs/RAGE axis induced NFκB activation. The AGEs/RAGE/NFκB-driven sustained inflammation and oxidative stress activation of signaling cascades play an important role in the pathogenesis of diabetic complications. AGE: advanced glycation end products; IL-1β: interleukin 1 beta; IL-6: interleukin 6; TNFα: tumor necrosis factor alpha; COX-2: cyclooxygenase-2; p27: cyclin-dependent kinase inhibitor 1B; TGFβ: transforming growth factor beta; NGF: nerve growth factor; ICAM-1: endothelial and leukocyte associated transmembrane protein; VCAM-1: vascular cell adhesion molecule 1; MCP-1: monocyte chemoattractant protein-1; MMP2: matrix metallopeptidase 2; ECM: extracellular matrix accumulation; IKβ: inhibitor of NFκB; IKK: inhibitor of IKβ; NFκB: nuclear factor kappa B; MAPK: mitogen-activated protein kinases; ERK: extracellular signal-regulated kinases; CREB: cAMP-response element binding protein; PKC: protein kinase C; JNK: Jun *N*-terminal kinase; AP-1: activator protein 1; JAK-STAT: janus kinase-signal transducer and activator of transcription.

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
