# Peer review of "Advanced Glycation End Products and Diabetes Mellitus: Mechanisms and Perspectives"

_biomolecules, 2022, doi:10.3390/biom12040542_

Round 1
Reviewer 1 Report
This review by Khalid et al provides a timely perspective summarizing many salient points regarding the role of AGEs in the pathophysiology of diabetes. In particular, the authors do a fine job of summarizing the interactions between AGE activation of RAGE, and downstream signaling pathways leading to inflammasome activation, ROS generation, mitochondrial dysfunction, and their roles in promoting and sustaining diabetic complications. However, there is some redundancy in the text which could be eliminated without sacrificing content, as well as some additional minor points which should be included. The following points should be addressed in a revised version.
- The endogenous generation of methylglyoxal via the triose phosphate isomerase reaction of glycolysis is an important source of AGEs. Figure 1 should be amended, and the text should be modified to reflect this.
- Some text and references regarding nucleic acid and lipid AGEs should be included since this is an emerging epigenetic modification of potential relevance to diabetic complications.
- The Future Directions section, particularly lines 366-382, contains some redundancy from previous sections on the AGE/RAGE axis. References should be folded into the earlier sections where AGE/RAGE signaling pathways are discussed (~ lines 226-244).
4. The potential therapeutic options for minimizing AGE induced inflammation and oxidative stress are appropriate for the Future Direction sections and should also include some of the pharmacological approaches to increase GLO1 expression, which have recently been reviewed (Biomedicine & Pharmacotherapy 131 (2020) 110663). The carbonyl scavenging properties of urea-based small molecules such as aminoguanidine and metformin should also be included as an approach to limit AGE formation.
Author Response
Response to Reviewer 1 Comments
This review by Khalid et al provides a timely perspective summarizing many salient points regarding the role of AGEs in the pathophysiology of diabetes. In particular, the authors do a fine job of summarizing the interactions between AGE activation of RAGE, and downstream signaling pathways leading to inflammasome activation, ROS generation, mitochondrial dysfunction, and their roles in promoting and sustaining diabetic complications. However, there is some redundancy in the text which could be eliminated without sacrificing content, as well as some additional minor points which should be included. The following points should be addressed in a revised version.
- The endogenous generation of methylglyoxal via the triose phosphate isomerase reaction of glycolysis is an important source of AGEs. Figure 1 should be amended, and the text should be modified to reflect this.
- Some text and references regarding nucleic acid and lipid AGEs should be included since this is an emerging epigenetic modification of potential relevance to diabetic complications.
- The Future Directions section, particularly lines 366-382, contains some redundancy from previous sections on the AGE/RAGE axis. References should be folded into the earlier sections where AGE/RAGE signaling pathways are discussed (~ lines 226-244).
- The potential therapeutic options for minimizing AGE induced inflammation and oxidative stress are appropriate for the Future Direction sections and should also include some of the pharmacological approaches to increase GLO1 expression, which have recently been reviewed (Biomedicine & Pharmacotherapy 131 (2020) 110663). The carbonyl scavenging properties of urea-based small molecules such as aminoguanidine and metformin should also be included as an approach to limit AGE formation.
Response
Thank you for your review. You have raised many important points here. We agree with all of your comments. Therefore, we tried our level best to follow your suggestions.
- We have included the endogenous generation of methylglyoxal via the triosephosphate isomerase reaction of glycolysis in the text and amended Figure 1.
- The lines 121-126 include the text and references regarding advanced lipid end products (ALEs).
- References have been folded into the AGE/RAGE signaling pathways section.
- In the future direction, some of the pharmacological approaches to increase GLO1 expression have been included. The carbonyl scavenging properties of urea-based small molecules have been also added to the manuscript.
Reviewer 2 Report
Comments on manuscript Biomolecules-1619928
entitled
“Advanced glycation end products and diabetes mellitus: mechanisms and perspectives” authored by Khalid et al..
In the manuscript the authors review the genesis of advanced glycation end products (AGEs) and their many roles in the etiology of diabetes mellitus type 2. Due to the broad interest and the numerous studies already published on this topic, it is no easy task to provide a comprehensive review, which covers “everything” and which is also easy to read.
The manuscript would benefit a lot from omitting the redundant descriptions, for example of actions of AGEs and mechanisms of RAGE-signalling. These could be described in detail once, and subsequently only the tissue-specific deviations from the general mechanism would have to be described (for example, lines 104-122 and 483-499 are widely redundant).
The many redundancies make it hard to easily read the manuscript and to get the point of interest of each paragraph/section.
It seems that a lot of the references are review articles themselves. To provide clarity, where original research and where reviews are cited, respectively, the authors could “indicate” cited reviews (for example in a way like this: …..text (ref A, ref B, and also reviewed in ref C)).
The text and also the figures contain numerous typos, and some sentences seem to be not complete. Please, control carefully.
Author Response
Reviewer 2
In the manuscript the authors review the genesis of advanced glycation end products (AGEs) and their many roles in the etiology of diabetes mellitus type 2. Due to the broad interest and the numerous studies already published on this topic, it is no easy task to provide a comprehensive review, which covers “everything” and which is also easy to read.
The manuscript would benefit a lot from omitting the redundant descriptions, for example of actions of AGEs and mechanisms of RAGE-signalling. These could be described in detail once, and subsequently only the tissue-specific deviations from the general mechanism would have to be described (for example, lines 104-122 and 483-499 are widely redundant).
The many redundancies make it hard to easily read the manuscript and to get the point of interest of each paragraph/section.
It seems that a lot of the references are review articles themselves. To provide clarity, where original research and where reviews are cited, respectively, the authors could “indicate” cited reviews (for example in a way like this: …..text (ref A, ref B, and also reviewed in ref C)).
The text and also the figures contain numerous typos, and some sentences seem to be not complete. Please, control carefully.
Response
Thank you for your review. You have raised many important points here. We agree with all of your comments. Therefore, we tried our level best to follow your suggestions.
- The lines 104-122 describe the process of endogenous AGEs formation in the main text of the article while the lines 483-499 are from the Figure 1 legend related to the same text where we have explained in detail different pathways of endogenous AGEs formation. We have modified the text in the figure legend.
- Most of the review article references have been revised.
- Text and Figure typos have been corrected.
Reviewer 3 Report
The focus of this review is to summarize the role of the AGEs/RAGE axis in the pathogenesis of type 2 diabetes mellitus and its associated complications. Furthermore, it has presented an overview of future perspectives to offer new therapeutic interventions.
It is a comprehensive review. Figures and references are adequate.
I have found this paper relevant to the field of this journal. I have only one minor comment.
Minor point:
- Please, add a table for better overview of candidate molecules for new therapeutic interventions (together with their function in the signaling pathways).
- The article should be ended at the line 482. Following text is redundant. For this reason, remove the paragraph between the lines 483-499 and incorporate it into the chapter Sources of AGEs (paragraph between the lines 104-120). It would be more logic, since this paragraph describes Figure 1. Other two last paragraphs can be omitted.
- The titles of Figures 2 and 3 should include “RAGE” in the capital letters and the beginning of word “diabetic” in Figure 3 should be in the small letter.
I recommend this paper for acceptation after minor revision in the journal.
Author Response
Reviewer 3
The focus of this review is to summarize the role of the AGEs/RAGE axis in the pathogenesis of type 2 diabetes mellitus and its associated complications. Furthermore, it has presented an overview of future perspectives to offer new therapeutic interventions.
It is a comprehensive review. Figures and references are adequate.
I have found this paper relevant to the field of this journal. I have only one minor comment.
Minor point:
- Please, add a table for better overview of candidate molecules for new therapeutic interventions (together with their function in the signaling pathways).
- The article should be ended at the line 482. Following text is redundant. For this reason, remove the paragraph between the lines 483-499 and incorporate it into the chapter Sources of AGEs (paragraph between the lines 104-120). It would be more logic, since this paragraph describes Figure 1. Other two last paragraphs can be omitted.
- The titles of Figures 2 and 3 should include “RAGE” in the capital letters and the beginning of word “diabetic” in Figure 3 should be in the small letter.
I recommend this paper for acceptation after minor revision in the journal
Response
Thank you for your review. You have raised many important points here. We agree with all of your comments. Therefore, we tried our level best to follow your suggestions.
- The chapter future perspective includes the overview of present and future candidate molecules along their mechanism of action to counteract AGEs/RAGE pathway. We have added some additional points in same chapter in the revised version. We have plans for another review that will only concentrate on the detailed description of candidate molecules for new therapeutic interventions. The table will be included in that.
- The article ends at line 482. The lines onward 483 are included in Figure 1 legend. Due to redundancy in the text, we have omitted one paragraph from Figure 1 legend.
- We have corrected the titles of Figure 2 and 3 as per your suggestion.
Round 2
Reviewer 2 Report
The revised manuscript is improved at several points, but there are still some points to be corrected.
Line 459: what does prodsource mean?
General: It would be helpful to differentiate between proteins (PROTEIN) and the genes encoding them (GENE)(italics) according to HUGO-nomenclature rules.
Author Response
The revised manuscript is improved at several points, but there are still some points to be corrected.
Line 459: what does prodsource mean?
General: It would be helpful to differentiate between proteins (PROTEIN) and the genes encoding them (GENE)(italics) according to HUGO-nomenclature rules.
Response
Thank you for your review. The typo error ‘prodsource’ has been corrected.
All the gene symbols are italicized according to HUGO-nomenclature rules (See manuscript).